# Automated Multi-Level Dynamic System Topology Design Synthesis

**Aart-Jan Kort [1], Jan Wijkniet [2] , Alexander Serebrenik [3] and Theo Hofman [4,*]**

1   Freelance Software Developer, 4261 TL Wijk en Aalborg, The Netherlands; aart-jan@kortictentechniek.nl
2   Punch Powertrain, 5653 LD Eindhoven, The Netherlands; jan.wijkniet@punchpowertrain.com
3   Software Engineering and Technology Group, Department of Mathematics and Computer Science,
    Eindhoven University of Technology, 5600 MB Eindhoven, The Netherlands; a.serebrenik@tue.nl
4   Control Systems Technology Group, Department of Mechanical Engineering,
    Eindhoven University of Technology, 5600 MB Eindhoven, The Netherlands
*   Correspondence: t.hofman@tue.nl; Tel.: +31-40-247-2827

**Abstract:** Designing new mechatronic systems for vehicle applications is a complex and time-consuming process. The increasing computational power allows us to generate automatically novel and new mechatronic discrete-topology concepts in an efficient manner. Using state-of-the-art computational design synthesis techniques assures that the complete search space, given a finite set of system elements, is processed to find all feasible topologies. The topology generation is done by converting the design synthesis problem into a constraint satisfaction problem. Accordingly, this mathematical problem is solved by assigning the presence of components and connections to variables, whereby a set of mathematical constraints need to be satisfied. These constraints capture, in essence, formalized engineering knowledge. After solving this problem, the results are post-processed to discard redundant topologies due to isomorphism. In this paper, a newly developed software application with automated constraint generation is presented that facilitates the topology generation with multiple system levels in a loop. The scalability of the problem and the different levels of expressiveness are analyzed, and the influence of the abstraction level choice on the search space is discussed. Finally, a relevant mechatronic design study from the automotive engineering field is discussed concerning the topology synthesis of alternative electro-hydraulic actuation systems being part of new continuously variable transmission topologies, thus showing its applicability.

**Keywords:** generative engineering; computational design synthesis; constraint programming; discrete topology design; mechatronic systems; mechanical engineering; continuously variable transmissions; vehicle technology

## 1. Introduction

Mechatronic system design is a complex and time-consuming job. For example, a conventional vehicles equipped with an engine and a robotized transmission with increasing integration of electronic control units have been built over decades now. As a result, a knowledge base is achieved about how to build such systems. With the introduction of advanced new technologies, like electric-drive systems and advanced hybrid power-split configurations, there is a lack of experience in designing such novel systems. To solve this problem, computational design synthesis (CDS) is one of the techniques that can be used effectively, and it focuses on automating synthesis during the design process. Through the support of (increasing) computation power, the drafting and structural analysis become easy tasks that can be outsourced by an engineer to a computer. A computer performs well on time-consuming or

tedious tasks. However, the user should define the problem and must interpret the results cautiously. With the use of CDS, the creative phase of designing a product is supported by software as well.

## 1.1. Automated Computational Design Synthesis Methods of Discrete System Topologies

Constraint programming (CP) techniques are already used to generate hybrid electric vehicle (HEV) topologies in [1] or for the optimization of mechanical systems [2]. In these cases, CP is introduced because of the system complexity and the multi-objective character, i.e., the system cost, emissions, powertrain efficiency, and vehicle performance. Therefore, two types of constraints are defined in [1] to prevent infeasible topologies from being generated. The first category of functional constraints is introduced for a proper system behavior. The second category is the cost constraint type. This latter type is used to prevent, e.g., a redundant series connection between two or more identical components. This additional component does not add functionality, yet increases the system cost and complexity.

In [3], a generic framework and guidelines for CDS are proposed. Thereby, four major CDS activities are distinguished: representation, generation, evaluation, and guidance. At the beginning of the process, the representation is defined (step 1). After that, an initial design is generated (step 2) and evaluated (step 3). This evaluation is used as guidance (step 4) in the next design generation. These four steps are used in the search process for new designs. Basically, that work introduces the application of engineering knowledge into the generation step. In [4], constraint-based methods are listed for automated CDS of solution spaces. CDS is related to introducing artificial intelligence (AI) into the design process of mechanical topologies.

For manufacturing and assemblage of mechanical parts, the use of a computer is already standard. Software can convert technical drawings into a program for a milling machine, and robots are able to assemble parts without the control of humans. However, when the product design is finished and already in production, then the costs are very high to change the (architectural) design. Ideally, every possible topology should be simulated and evaluated in an early design stage. This reduces the risk of having selected a suboptimal architectural solution. In [5–14], the importance of introducing computational power into the design process due to the increasing system complexity is emphasized.

The method in [5] is based on object-oriented graph grammars. That method uses a hybrid knowledge representation to formulate generic design rules in graph grammar. In this graph grammar, every function, behavior, and structure (FBS) is defined for a particular problem. By means of an open-source software application, topologies are automatically generated based on the formalized knowledge in the graph grammars. In [8], the graph-theoretic method is used to generate designs. The authors of [5] strongly recommend new supporting efforts from the field of computer science for rapid progress in this area. Recently, a generic CDS framework has been modified and tailored to complex dynamical systems in [9], with an example case study on advanced continuously variable transmissions (CVTs) for vehicles. Thereby, CDS is applied on a lower system level compared to [1]. In [1], the search space and constraints are manually defined to generate feasible designs, while in [9], the dynamic models of feasible topologies are automatically composed and simulated in Matlab Simscape® for automated evaluation purposes, which is seen as a significant improvement.

## 1.2. Research Contributions and Outline

Based on the literature review above, it is shown that the research area of CDS for active dynamic systems is becoming increasingly more relevant. Therefore, the work of [1,9] in particular is here further extended with:

(a) A CDS framework to automatically formalize engineering knowledge into generic constraints. Moreover, this enables the automated generation of multi-level systems with predefined constraints;

(b) The generation of multi-domain (electrical, mechanical, hydraulic, or combinations thereof) discrete system topologies. This requires a library format that specifies the domain of each component. For example, an electric machine has an electrical input and a (rotational) mechanical

output. Based on these assigned domains, the constraints on the connection variables can automatically be derived;

(c) The generation of topologies with different levels of expressiveness. Whereas [1,9] only declares component types and instances, this work introduces the declaration of ports and analyzes the benefits and drawbacks of this higher level of expressiveness. Furthermore, this higher level of expressiveness supports (b);

(d) Further analysis of the search space: The influence of the number and degree of components vs. computational time as well as the working principle of the solving algorithm is analyzed and insights are created.

The research objectives (a) and (b) are considered as future work in [5], and follow the trend towards a complete product design using CDS. Given (b), a library format is required in order to define all components' properties, including the energy domain. Objective (b) transforms the topology generation problem into a port-matching problem, which is, however, still interpreted as a satisfiability problem. In this suggested library format, all the components are specified with their corresponding ports. When a component is subject to an energy conversion, like from electrical to mechanical, then the ports of the component must be properly connected to ensure proper functionality. For the intended analyses, as described in (c) and (d):

- A new computer-aided engineering (CAE) software tool has been developed, enabling the user (system engineer) to quickly set up and solve CSPs from the system to the component level, fully automated, in the field of powertrain systems.

Hence, different CSPs can be put in a queue and solved successively for analysis purposes. Moreover, to obtain deeper insights into the topology design synthesis problem, the following aspects are also considered in this work: the size of the search space and the computation time to solve the constraint satisfaction problem (CSP). A second technological innovation and result from the tool applied in a relevant case study comprises:

- New electro-hydraulic actuation systems to be used in a continuously variable transmission (CVT) system (friction-based transmission type, cf. Figure 1); this is selected as a representative topology synthesis case study due to the high level of complexity and in order to demonstrate the multi-domain capabilities of the software application.

This subsystem type covers multiple domains; e.g., electrical (machines), mechanical (pulleys, gears), and hydraulic (pump, valve) components are, typically, used to construct such designs. Next, the multi-level capabilities are evaluated with use of the CVT topologies generated in [9]. Finally, this case study, elaborated upon in Section 7, shows a design space of $> O(10^{17})$ possible topologies using a library that contains 14 components.

The outline is as follows: Firstly, this paper introduces the developed methodology in Section 2. Next, in Section 3, the CSP programming and solving are further explained. The work is continued with Section 4 about the automatic constraint generation and the implemented constraints. Furthermore, the scalability of the method is considered in Section 5. The abstraction level choice of the topologies is described in Section 5.5. The following section, Section 6, discusses the possibility of multi-level topology generation. The research is finalized by applying the method to a case study, described in Section 7. Finally, the conclusions and recommendations for future research are described in Section 8.

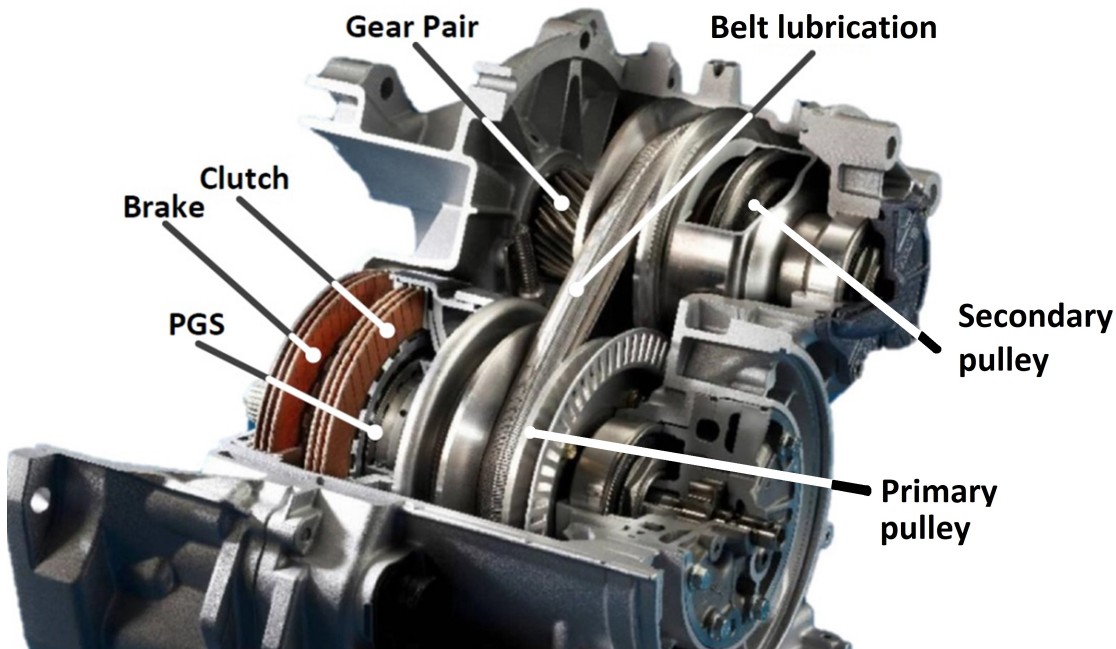

**Figure 1.** Continuously variable transmission (CVT) technology (courtesy of Punch Powertrain).

## 2. Topology Generation Methodology

Here, the developed methodology in this research is introduced. One of main goals of this research (cf. (a) in Section 1.2) is to automate the generation of topologies, starting with the use of formalized engineering knowledge and a library of components selected by a system engineer. This process is visualized in Figure 2.

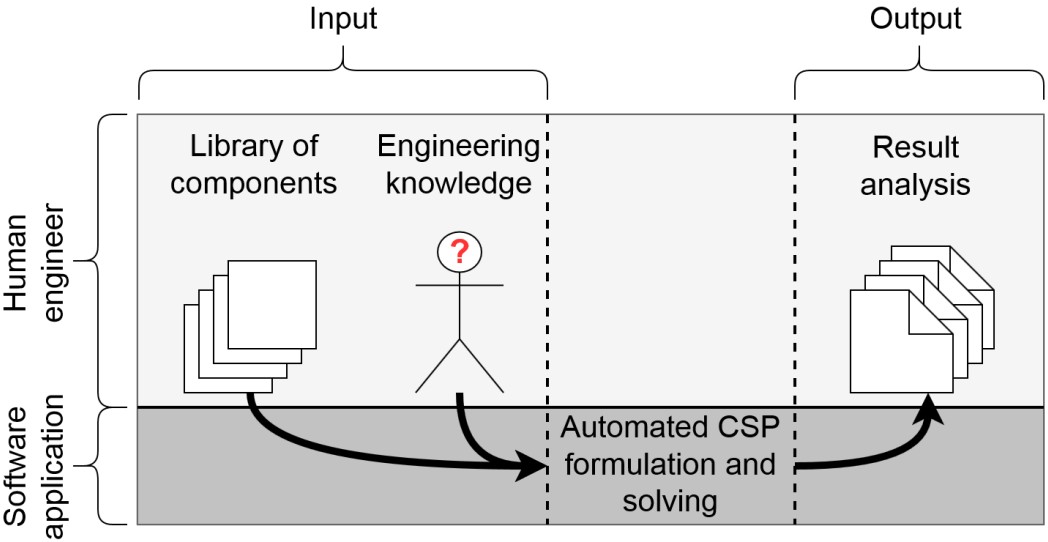

**Figure 2.** Visualization of methodology: generation of topologies with the use of formalized component properties and engineering knowledge.

To support this research, a software application has been developed that formulates and solves the constraint satisfaction problem (CSP). First, the components and their properties are stored in a session library. Secondly, the engineering knowledge is formalized and stored. The next step is applying the formalized knowledge to the session library and expressing it mathematically. As a result, the problem of topology generation is converted into a CSP. This set of mathematical and logic

expressions is solved and converted into easily interpretable graphs (topologies). Finally, the results can be analyzed by a system engineer. The five discrete steps of this methodology are:

Step 1:    Creation of a library;
Step 2:    Formalization of engineering knowledge;
Step 3:    Formulation and solving of a CSP;
Step 4:    Result post-processing;
Step 5:    Result analysis and classification.

Below, the methodology is sequentially discussed.

### 2.1. Step 1: Creation of a Library: Identification of Useful Components

First, a library of components and a finite number of input and output ports for each of these components are defined and specified by the user. The number of ports represents the degree of the component. The following generic properties are assigned to each port: energy domain, flow direction, and controllability. In the developed method, two different libraries can be distinguished: a base library and a session library. The base library contains unique (hydraulic) components, e.g., *Pump*, *Sump*, and *Valve*. When a new project is created, e.g., the topology generation for an electro-hydraulic actuation system, a session library is composed. This basically assigns the instances to each component from the base library.

### 2.2. Step 2: Formalization of Engineering Knowledge: Mapping Functions to Components

A second requirement is the formalization of the engineering knowledge and experience into mathematical constraints by mapping functions to components. During the generation of feasible topologies, constraints are used to determine whether a topology is feasible. The feasibility of a topology depends on several design considerations; for example:

- Are the components correctly physically connected? For example, are the components from different domains properly interconnected?
- Does a topology contain any unconnected components? For example, are all the ports of each component connected?
- Does the topology meet the functional requirements? For example, to power a hydraulic pump, rotational energy needs to be supplied by an engine or electric machine.

The quality of the generated (feasible) topologies depends on the completeness of the set of the constraints. This completeness is related to the number of infeasible topologies as part of the total generated topologies. The more complete the set is, the fewer infeasible topologies are prevented from being generated. However, the constraint set can also be too strict (overregulation). As a result, feasible topologies are prevented from being generated due to carelessly defined constraints.

Constraint Classification

To make the set of constraints as complete as possible, two subsets are introduced: general and custom constraints. General constraints are applied to all components; for example, a constraint to prevent redundant topologies with only the identifier is interchanged between those topologies. These are hard coded and can be evaluated by a system engineer easily. The input of engineering knowledge can be defined with the use of custom constraints. The most fundamental constraint in the generation of topologies is whether a component is present and that the number of connections with this component is equal to the number of ports.

The constraints can be further classified into four different constraints: functional, redundancy, physical, and application-specific [9]. Functional constraints are needed to ensure the required functionality in the system. Next, there are physical constraints. This class is used to prevent, e.g., the connection between a battery cell and a hydraulic valve, since their energy domains do

not match. A constraint of the redundant class can prevent a sequence of two identical hydraulic valves or two identical clutches. These sequences do not add functionality, yet raise the cost and complexity. Application-specific constraints are specific to the project. For example, the technology choice can be defined with such a constraint; for instance, when hydraulic topologies with five pumps, or transmission topologies with or without a planetary gear set must be generated. The presence of a component can be forced or prevented with constraints.

### 2.3. Step 3: Formulation and Solving of a CSP: Using SWI-Prolog

The system topology generation, which satisfies a set of constraints, is performed with the use of a CSP solver. The mathematical problem consists of tuple variables, domains, and constraints. As described in [1], constraint programming (CP) is suitable for tightly constrained problems. A formal description of this mathematical problem is given by a set [1]:

$$CSP = \langle \mathbf{X}, \mathbf{D}, \mathbf{C} \rangle \tag{1}$$

$$\text{with: } \mathbf{X} = \{X_1, X_2, \ldots, X_n\}, \tag{2}$$

$$\mathbf{D} = \{D_1, D_2, \ldots, D_n\}, \tag{3}$$

$$\forall \, i \in \{1, 2, \ldots, n\}, X_i \in d_i = \{0, 1\}, \tag{4}$$

$$\mathbf{C} = \{C_1, C_2, \ldots, C_m\}. \tag{5}$$

The finite set of variables $\mathbf{X}$ represents two aspects: the individual components (as vertices in a graph) from the session library and the connections between the components (as edges in a graph). Here, the components are identified with characters. Therefore, for each component in the session library, a character is assigned to the component, such as $A, B, \ldots, Z$. If the session library contains more than 26 components, two-character identifiers can be used, such as $AA, AB, \ldots, ZZ$. The connections between the components are represented by the identifiers from both components. Variables representing the connections between components are declared once in alphabetical order. Connections are considered as unidirectional; therefore, the connections $AB$ and $BA$ are represented by $AB$. Subsequently, when two-character identifiers are used for components, the connection identifier has four characters. The domain $\mathbf{D}$ for every variable (components and/or connections) is 1 (present) or 0 (not present). A set of constraints $\mathbf{C}$ completes this type of mathematical problem. These constraints are used to formalize engineering knowledge and design experience from Step 2.

Constraint Logic Programming over Finite Domains

The CSP is solved with SWI-Prolog [15], including the constraint logic programming (CLP) over finite domains (FDs) library [16]. This combination is selected because of the following benefits in comparison to other solvers, such as GNU Prolog, B-Prolog, or ECLiPSe. In relation with GNU Prolog, the CLP(FD) library is more consistent when it exceeds the predefined limits. Another benefit is that SWI-Prolog can be controlled by another program, which is useful for automating the topology generation process. Next, the software is freely available and open source. Finally, the functions implemented in the CLP(FD) library are suitable for formalizing engineering knowledge into constraints. Two major aspects of a solver are: firstly, the expressiveness of the used language—for example, the number of lines needed to express the formalized engineering knowledge— and secondly, the solver speed. In general, the lower the expressiveness level, the faster the solving.

In Section 3, the notation and solving algorithm used will be further elaborated upon. Future research may include a comparative study of different solvers for the application of discrete topology generation, since this is beyond the scope of the current research.

### 2.4. Step 4: Result Post-Processing: Using Filter Designs

After solving the CSP, the set of topologies is post-processed by applying different filters. These filters (seen as post-processing constraints) may be developed due to a tradeoff between the initially required (and perhaps possible) level of expressiveness and the overall computation time. For example, when a group of components is not connected to the main power flow of the topology, i.e., to any 'power source' or 'power sink', these topologies are redundant and need to be discarded. With this (quickly evaluating) filter, there are two options. First, it is only checked whether there is a (power) path from a sink to a source. Second, a stricter (and slower) filter requires that every component is connected to a source. This optional filter choice depends on the application of the concepts. For example, as discussed in our case study later on, a transmission (subsystem level) should not have separated power paths, unlike a hydraulic system (component level).

Isomorphism Detection

The second post-process filter comprises isomorphism detection and removal. With the introduction of virtual nodes (VNs), the topology isomorphism becomes an issue. These node types were introduced in [1] in order to connect three components properly; they have no key properties assigned to them. Hence, when two VNs are connected in series and four components are connected, then the total number of different configurations is 24 at the port level of expressiveness (cf. Figure 3).

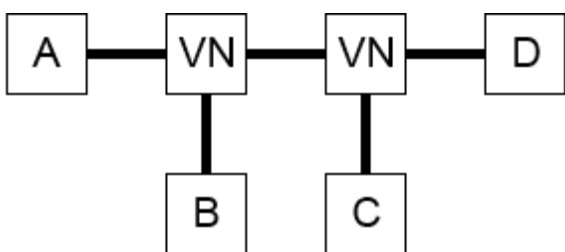

**Figure 3.** Illustration of an isomorphism topology that includes two virtual nodes (VNs) and four first-degree components (A–D). This results in 24 different possible configurations, of which 23 are isomorphous.

This can be calculated with $k$ permutations of $n$ components, where $k = n$ and the number of topologies becomes $n!$ or, hence in this case: $4! = 24$. Consequently, when topologies are generated, then only one topology is valid, and the other 23 are isomorphous to the first one and are discarded from the set of generated topologies.

This phenomenon of isomorphism is prevented by removing the VNs from the string that contains all connections of a topology and replacing them with all the possible connections between the components that the VN connects. This solution procedure is visualized in Figure 4. Next, isomorphism can easily be detected through a piecewise topology comparison.

Another cause of isomorphism is when there multiple instances of the same components are present. In Figure 5, two isomorphous topologies are shown due to such multiple instances, since $B_1$ and $B_2$ are of the same component type, so topologies $X$ and $Y$ are isomorphous.

It is assumed here that the components are bidirectional. Due to the raised abstraction level, the solver cannot detect that they are identical, since their identifiers are different. This is solved by editing the connection string, which contains every connection in the topology. When the session library contains more than one instance of a component, all identifiers representing that particular component are replaced by the first instance. This is done for all topologies. These edited connection

strings are compared to each other to find the isomorphous topologies. Post-processing the results is not preferred, since it is relatively slow compared to the CSP solving; however, it is considered unavoidable due to the inherent nature of assigning the identifiers to the components.

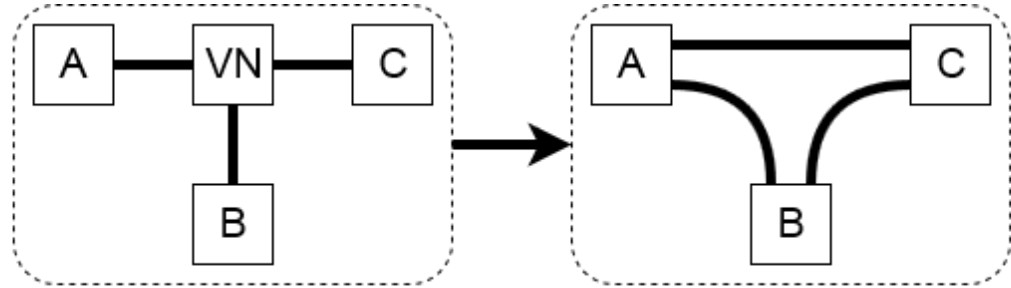

**Figure 4.** Remapping the connections to discard isomorphous topologies due to virtual nodes (VNs).

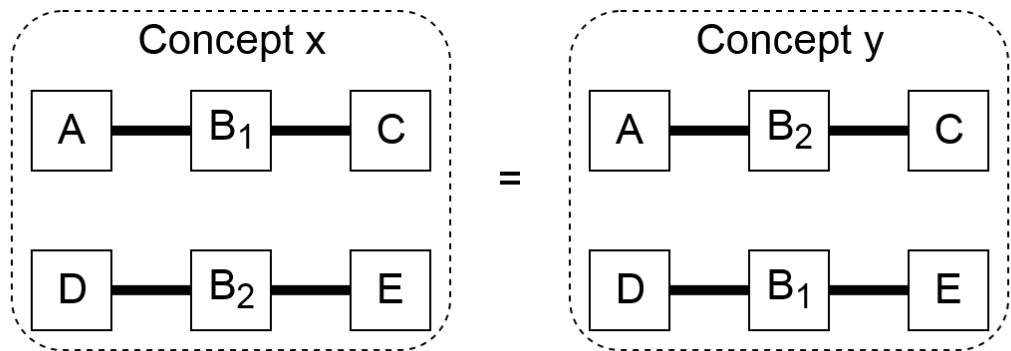

**Figure 5.** Two isomorphous topologies introduced by different component identifiers for multiple instances of a component.

*2.5. Step 5: Result Analysis and Classification*

If the generation phase is finished, then the topologies can be classified, for example, based on the present component instances. As a result, the size of the design space can be reduced or analyzed with respect to the distribution of the different topologies. Additionally, based on the gained results, new constraints can be defined and implemented. The process of refining the constraint set and analyzing the results is an iterative process. The analysis of the generated topologies can be done with the application's built-in result viewer. This viewer provides a graphical representation of the generated topology design. When this is finished, then the (dynamic) system performances (e.g., efficiency, top speed of the vehicle, ratio tracking accuracy, etc.) of the static topology models can be evaluated by an automated parameterized modeling step (e.g., using Matlab or Simulink) that transforms each relevant topology into a scalable dynamic system topology design analysis model. Note that this model-based evaluation step can then be used as well to classify topologies, e.g., based on their operation modes [9]. This evaluation step is not discussed further in this work.

**3. CSP Programming and Solving**

As mentioned in Section 2.3, the process of formulating and solving the CSP is here further elaborated upon using illustrative examples. The constraints of the CSP problem are programmed and solved with SWI-Prolog. For every combination of the variable values, the set of constraints is checked to determine if the whole set is true. If the set is true, then every label of the variables with the value of one (component and/or connections) is written to an output file. The solving algorithm and the suggested and implemented solving improvements are explained in Sections 3.2 and 3.3, respectively.

Here, mainly two categories of constraints are used from the CLP(FD) library to formalize engineering experience and knowledge: arithmetic and reification constraints, as discussed below.

### 3.1. Arithmetic and Reification Constraints

The arithmetic constraints are used to compare two expressions. For every expression, the equality or inequality is checked. Such an expression can be a constant or a result of addition, subtraction, or multiplication. Furthermore, maximum and minimum functions are supported. In addition, constraints can be defined with a condition, such as the presence of one or more connections and/or the presence of a certain component. A couple of the constraint types used are defined in Prolog code and provided in Listing 1; each constraint is discussed below.

Explanation of constraints (1)–(7):

(1) These constraints use a logical equality. The truth table of logical equality is shown in Table 1. The only solution for this constraint is that A is always present.
(2) In the definition of the search space, this constraint is used to force the presence of a component (1) or to prevent a connection (2) from being generated.
(3) This constraint combines a sum, a logical operator, and a relational operator. This results in a constraint that limits (due to the relational operator) the presence of components A, B, and C to a maximum of two out of three.
(4) This constraint uses classic negation, material implication, and an AND logic connector. This is to prevent connection CD (due to the negation) when (due to the material implication) there are connections AB and (due to the AND gate) BC. The truth table of the AND, OR, and NOT logic is shown in Table 2.
(5) This constraint is the same as (4), except for the OR gate instead of the AND gate. Now, connection AC is prevented when there is a connection AB or (due to the AND gate) CD.
(6) This combines the OR and AND gates from (4) and (5), respectively, to prevent connection BD.
(7) This constraint combines a sum function to force the number of connections of D. For example, component D is a second-degree component, i.e., it can have two connections (two edges). To force that this component has two connections, this constraint can be added. When component D is present, then there can also exist two connections with D. However, when D is not present (D = 0), there are no connections containing D.

Listing 1: Examples of constraint types in Prolog code that are used to convert engineering knowledge into a constraint satisfaction problem (CSP). Note that the '%' sign indicates a comment.

```
Components = [A,B,C,D],
ComponentsDomain 0..1,
Connections = [AB,AC,AD,BC,BD,CD],
ConnectionsDomain 0..1,
% 0 = not present
% 1 = present

A #<==> 1,                          %(1)
AB #<==> 0,                         %(2)
sum([A,B,C], #=<, 2),              %(3)
AB #∧ BC #==> #\ CD,               %(4)
AB #∨ CD #==> #\ AC,               %(5)
(AB #∨ AD) #∧ (AC #∨ BC) #==> #\ BD,%(6)
sum([AD,BD,CD], #=, 2*D),         %(7)
```

**Table 1.** Truth table for the material implications (⇒) and logical equality (⇔). Note: # indicates a Prolog notation.

| Input A | Input B | A ⇒ B | A ⇔ B |
|---|---|---|---|
| | | A # ==> B | A # <==> B |
| False | False | True | True |
| True | False | False | False |
| False | True | True | False |
| True | True | True | True |

**Table 2.** Truth table of the AND gate, OR gate, and NOT gate. Note: # indicates a Prolog notation.

| Input A | Input B | A AND B | A OR B | NOT A |
|---|---|---|---|---|
| | | A#∧B | A#∨B | #\A |
| 0 | 0 | False | False | True |
| 1 | 0 | False | True | False |
| 0 | 1 | False | True | |
| 1 | 1 | True | True | |

## 3.2. SWI-Prolog Solving Algorithm

Prolog is a logic programming language that is widely used in scheduling, sequencing, and routing problems [17]. Here, the working of constraint solving is described. The search algorithm used by Prolog is a backtracking tree search extended with local consistency checking and constraint propagation [18]. An example of a backtracking tree search is illustrated in Figure 6.

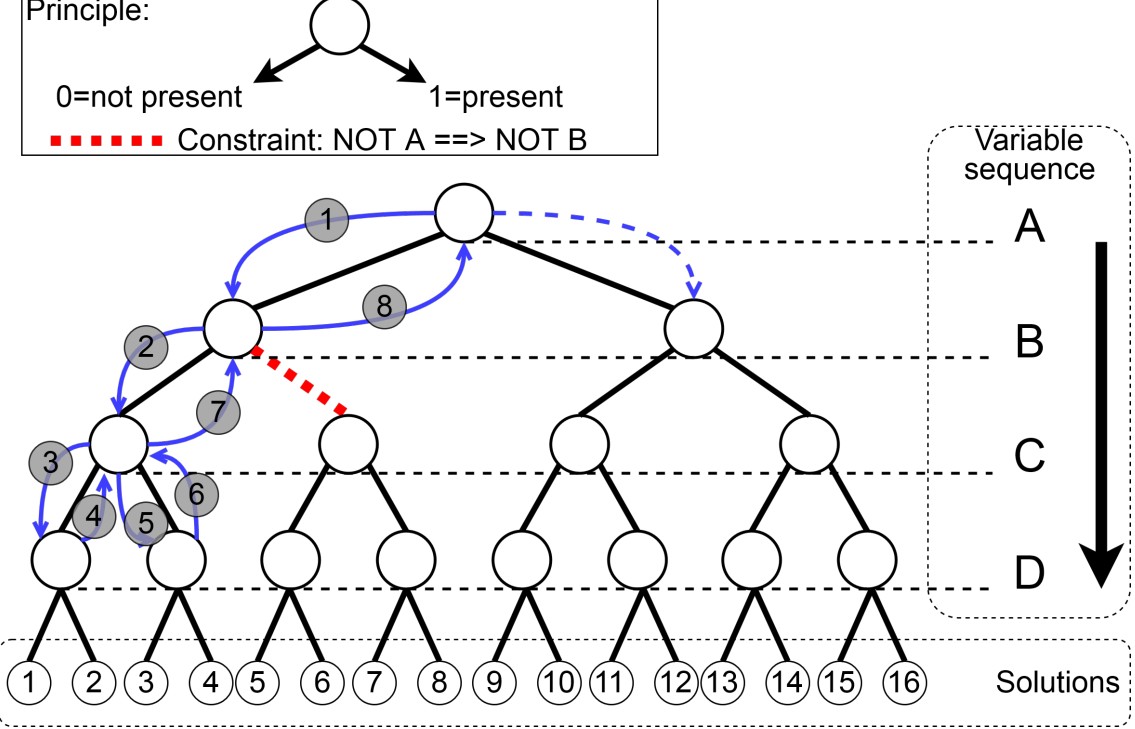

**Figure 6.** Illustration of the first eight steps of searching with a depth-first search and backtracking. Solutions are listed in Table 3.

**Table 3.** Solutions for the search algorithm example depicted in Figure 6.

| Solution # | | Solution # | |
|---|---|---|---|
| 1 | - | 11 | AC |
| 2 | D | 12 | ACD |
| 3 | C | 13 | AB |
| 4 | CD | 14 | ABD |
| 9 | A | 15 | ABC |
| 10 | AD | 16 | ABCD |

As can be seen, after step three, the solutions one and two are generated, i.e., the constraint set is satisfied, and the variables, which are one, are written to the results file. In this figure, the search steps and solutions are displayed. The displayed red-colored 'path' is conditionally constrained with $NOT\ A \Rightarrow NOT\ B$. With the mentioned search extensions, the solver 'tries' to simplify the problem. As a result of that, the number of backtracking steps can be reduced significantly by filtering values from some domains. Due to this, after search step 7, the tree is not further searched with $A = 0$ and $B = 1$. Consequently, the solutions 5–8 are not possible. The solutions of this search are listed in Table 3. In the next section, the implementation of a different variable sequence selection is described, which results in better solving times.

*3.3. Improvements Implemented in the Solving Process: Reducing Computation Time*

First, the variable selection strategy can be changed into different approaches to speed up the solving process. In the example above, the variable selection is A, B, C, and D. The solver default is the order of variable definition in the script. The variable declaration of the components in the Prolog file is based on the component sequence in the base library. The solving strategy can be adjusted to the domain sizes. However, this is not advantageous, since the domain is the same for every variable. The preferred strategy is to select the variable that participates in the most constraints first. The solving process is about six times faster with this strategy compared to the default selection strategy. Note that the value order is, by default, ascending, as shown by the sequence listed in Table 3, and changing this to descending does not affect the solving time. Finally, the branching strategy can be set. This does affect which and how many values are chosen from the domain. This can be 'step', 'all', or the 'mean value' of the domain. This is kept as 'step' by default due to the chosen domain of binary values.

SAT Solvers

The solving time may also be reduced by selecting another solver. Consequently, the notation of the constraints can differ from the current notation. For instance, the 'sum' expression, which is part of the function set of the CLP(FD) library, might be rewritten into a set of logic expressions. Although the number of constraints increases, the solving time will be smaller due to the efficiency of the satisfiability problem (SAT) solvers. SAT solvers are able to process such significant constraint sets more efficiently. Basically, the constraint set becomes larger, yet the solving time is reduced simultaneously.

**4. Automated Constraint Generation**

In general, the more complete the constraint set for the CSP is, the better the results regarding the number of feasible topologies as part of the total number of topologies will be. To speed up the process of topology generation, the constraint generation should also be automated. This is done by applying the defined general and/or custom constraints to the session library. The more properties are defined for every component, the better the capabilities for automated constraint generation. Here, the set of constraints is applied to the session library. In the case that the library does not contain one or more individual components, the constraint is skipped. The power of automated generation is that all possible combinations of components are covered. This is explained by means of the notations

on three different (types of semantic) levels for an arbitrary trivial constraint using the components (variables) from a library: (1) engineering level, (2) mathematical level, and (3) logic level. Using a simple example, these levels are explained with the use of the following constraint defined at:

(1)   The engineering level:

"The output of an oil pump may not indirectly, by means of two virtual nodes, be connected to the input of an oil pump." Moreover, this constraint is visualized in Figure 7 and can be defined at:

(2)   The mathematical level:

$$(V_{6,i}, V_{4,k}) + (V_{4,k}, V_{6,j}) = 2 \Rightarrow (V_{6,i}, V_{6,j}) = 0$$

$$\forall \ i, j, k \in \{1, 2, 3\}, i \neq j,$$

where $V_{\tau,\iota}$ represents a node, with the component type $\tau$ and the instance of that particular component $\iota$ [1].

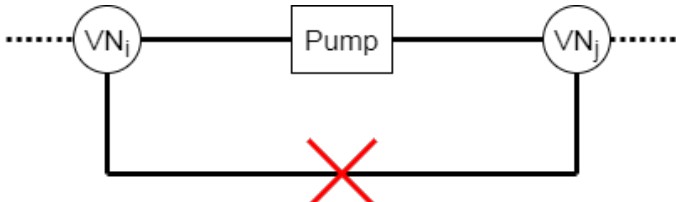

**Figure 7.** Visualization of the constraint: "The output of an oil pump may not indirectly, by means of two virtual nodes, be connected to the pumps input."

When the session library contains multiple instances of a component—for example, two electric machines and two virtual nodes—then this results in a constraint set that includes all possible sequences of these components. If, for example, the session library contains three instances of a pump (F, G, and H) and three instances of a virtual node that handles hydraulic energy (K, L, and M), then this results in the following expressions at:

(3)   The logic level:

$$(FK \wedge FL) \vee (GK \wedge GL) \vee (HK \wedge HL) \Rightarrow \backslash KL$$

$$(FK \wedge FM) \vee (GK \wedge GM) \vee (HK \wedge HM) \Rightarrow \backslash KM$$

$$(FL \wedge FM) \vee (GL \wedge GM) \vee (HL \wedge HM) \Rightarrow \backslash LM.$$

The strength of automated constraint generation also holds for preventing a loop between three virtual nodes. For instance, F, G, and H are the identifiers of virtual nodes and the constraint "When three virtual nodes are connected in series, the first and the last cannot connect" is applied, all possible sequences are covered, such as F-G-H, F-H-G, G-H-F, etc. Before the generation of constraints is done by the software application, the constraints can be set to as active or inactive. This enables the ability to do analysis on the influence of certain constraint(s).

## 5. Scalability of the Method

A major aspect of the proposed method is the scalability of the discrete topology generation problem: adding/removing components at single or over multiple system levels. Here, the limiting factor is the solving performance of the satisfiability problem. This is due to the software used (SWI-Prolog and the CLP(FD) library), which is, typically, running on only a single processor core. The search space of the problem can be defined as the number of variables that SWI-Prolog has

to process during the satisfiability check, since the domain size of each variable is two. The more components are in the session library and the more connections are possible between those components, the bigger the search space that needs to be processed is. The search space is, on a higher level, directly influenced by the chosen abstraction level, since this level does affect the number of components. The influence of the abstraction level choice is further discussed in Section 5.5. On a lower level, the search space is defined by the level of expressiveness, which will be explained below.

## 5.1. Influence of the Topology Expressiveness Level

There are two levels of expressiveness considered: the lower 'component' and the higher 'port' level, respectively. An example of these expressiveness levels is visualized in Figure 8 for an electro-hydraulic system.

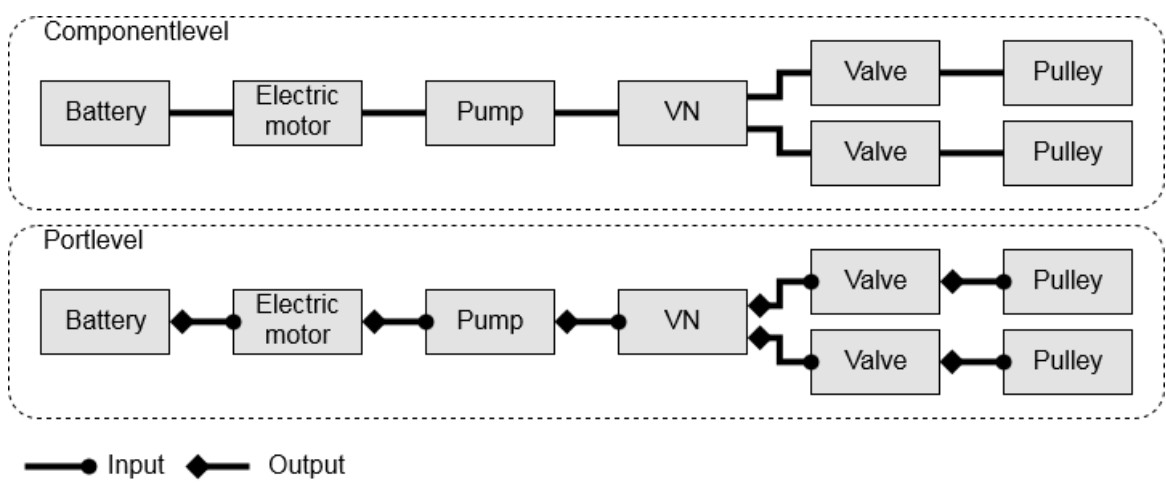

**Figure 8.** Example topology with two different levels of expressiveness.

Both levels have their advantages and disadvantages. The advantage of the component level is that the search space size is smaller than at the port level; hence, the time to solve is less, since fewer variables need to be processed while solving compared to the port level. The advantage of the port level is that topologies contain more details about the connections between the components. This can be useful, e.g., when Matlab models are generated from a topology's description. This becomes more relevant when components are multi-domain, i.e., the ports do not have the same energy domain (e.g., hydraulic, mechanical, or electrical). The port mapping of the topologies can be directly applied to the component connections of the model. The downside of the port level is that the search space increases due to all the possible connections between the ports.

In the following, the influence of the level of expressiveness on the topology generation is further quantified and evaluated. The solved example problem contains two energy sources, two converters, three virtual nodes, and three consumers (sinks). In Table 4, the influence of the chosen expressiveness level is quantified.

It can be observed that the size of the constraint set increases due to the higher number of possible connections between the components. The solving time is also higher due to the larger search space and the size of the constraint set. Moreover, topology isomorphism plays also an important role here, as stated in Section 2.4. At the port level, there are five times more topologies generated than at the component level. However, there are slightly more (unique) topologies generated. The difference in generated topologies at the component and port levels could be further investigated and could result in additional constraints.

The usage of these two expressiveness levels depends on the project phase and goals. In the beginning, when the search space is unclear and the constraint set incomplete, the lower component level is used, since it is faster. Accordingly, the generated topologies can be analyzed and more

constraints can be defined because of all of the discovered infeasible topologies. When this is finished, as a second step, the higher port level can be chosen. A higher expressiveness level offers more topology detail for multi-domain model-based evaluation. If components are multi-domain and/or have several ports, then the port mapping is more explicit and can be more robustly exported to, e.g., a Matlab Simulink environment.

**Table 4.** Influence of the expressiveness level on the Prolog script.

|  | Expressiveness Level | |
| --- | :---: | :---: |
|  | **Component** | **Port** |
| # Component variables | 11 | 11 |
| # Connection variables | 55 | 171 |
| # Constraints (characters) | 2K | 11K |
| # Total constraints | 26 | 168 |
| Search space size | $2^{62} \approx 5 \times 10^{18}$ | $2^{89} \approx 6 \times 10^{26}$ |
| Solving time (seconds) | 45 | 218 |
| # Raw solutions | 21,536 | 102,768 |
| # Post-processed solutions | 250 | 262 |

### 5.2. Component versus Port Expressiveness Level

For the constraint generation, every port combination is considered at the port expressiveness level; therefore, every port has a variable in the CSP. For every port pair, the properties are considered, and it is decided whether the connection between these ports must be prevented or not. When, for instance, the port energy does not match, the connection is prevented. To prevent a connection between two ports at the port level, one (Prolog) statement is needed, e.g., A1B1 #<==>0. At the component level, the constraint generation is more complex because of the lack of expressiveness. It is possible, yet more complex, with multi-domain components, like an electric machine or pump. When a session library contains multi-domain components, then, in order to generate the constraints and to split the energy domains for every connection, a common energy must be found. If this is not possible, then the components cannot be connected. When there is one or more common energy domain, then the components can connect and are added to a list. This is done for every distinct energy domain of a component. The sum of these connections should be equal to the number of ports that have that specific energy domain. Generally, at the port level, it is more straightforward to define the constraints due to the expressiveness compared to the component level.

### 5.3. Search Space Definition without Constraints

First, the search space size is considered for both expressiveness levels. The search space is the region where all solutions are located. The size of the search space indicates the number of variable combinations that the solver, like SWI-Prolog, has to process to get all solutions. Notice that the calculations below, used as a simple example, are performed here without any constraints. At the component level, the search space size, as the cardinality $|.|$ of the possible discrete topology set $\mathbf{T}^p$, can be quantified with:

$$|\mathbf{T}^p_{\text{component level}}| = 2^{\left(n_c + \frac{n_c \cdot (n_c - 1)}{2}\right)} \tag{6}$$

$$\text{with: } n_c = n_x + n_y + n_z, \tag{7}$$

where $n_c$ is the number of components. The notations $n_x$, $n_y$, and $n_z$ are, respectively, the numbers for the first-, second-, and third-degree components. The number of components is directly related to the number of connections, since every component is able to connect to every other component unless restricted by constraints. The search space on the port level depends on the number of components of every degree. The degrees of components are adopted from the graph theory, as applied in [1].

The degree of a component stands for the number of ports of a component, i.e., how many other components the particular component can connect to. Accordingly, the size of the search space on the port level can be calculated with:

$$|\mathbf{T}^p_{\text{port level}}| = 2^{\left(n_p + \frac{n_p \cdot (n_p - 1)}{2}\right)} \tag{8}$$

$$\text{with: } n_p = n_x + 2 \cdot n_y + 3 \cdot n_z, \tag{9}$$

with $n_p$ representing the total number of ports. It can be observed from the equations that the search space scales exponentially and with a factor of two. With the component expressiveness level, the increase is equal in every direction. For the port level, this differs; the higher the degree of the component, the higher the increase for every added component. That means that the number of second-degree nodes has a higher share in the total search space compared to first-degree nodes. In Figure 9, the solving time is shown as a function of the search space (measurements were performed using an Intel Core i5-M540 @ 2.53 GHz and 8 GB RAM with Win7 and SWI-Prolog 7.4.2.).

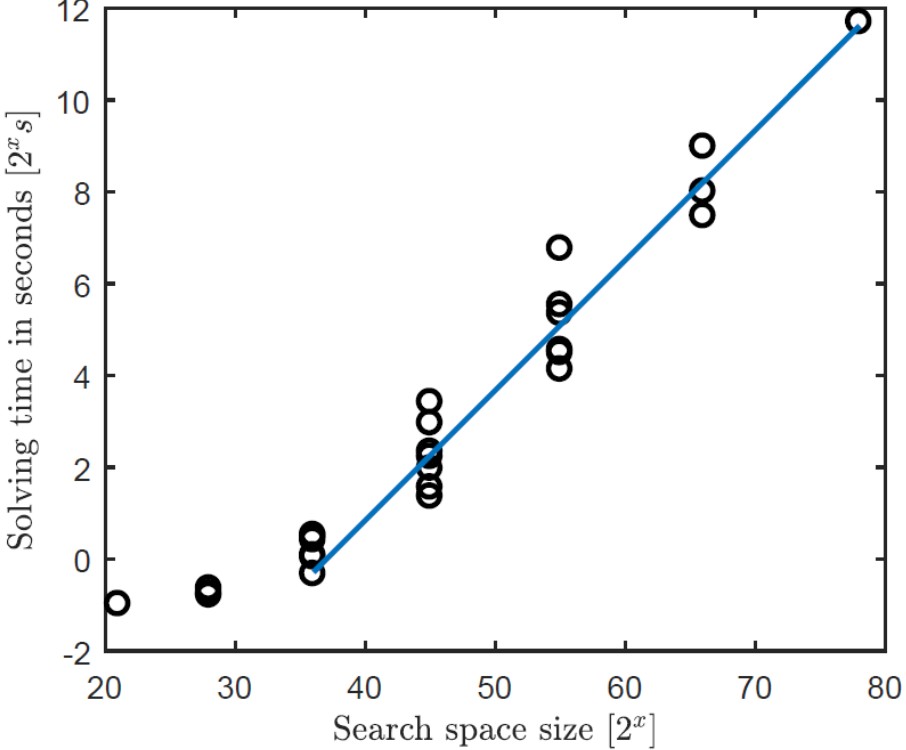

**Figure 9.** CSP solving time as a function of search space size. For the ease of reading: $2^{12}$ s ≈ 68 min.

When the search space is $<2^{40}$, then the software overhead of starting the solving process is more dominant than the solving time. When the search space is $>2^{40}$, then the solving time scales exponentially with the number of possible combinations. The influence of constraints on the solving time is considered in the next subsection.

## 5.4. Influence of the Constraints on the Solving Process

When an unconditional constraint is applied to one of the components and/or connections, the search space is affected instantly. This is probably due to compiler optimizations in SWI-Prolog. Conditional constraints do not instantly affect the search space. Therefore, all variables whereon conditional constraints are applied need to be processed during the satisfiability check. However, unconditional constraints do have an indirect influence on the process. This is due to

changing the variable selection strategy in the solver (see Section 3.2 for details). The more a variable participates in a constraint, the earlier this variable is processed by the searching algorithm. Depending on the defined constraints, this can lower the solving time by having fewer backtracking actions. Future research and measurements should be done on this topic to quantify this phenomenon more reliably.

## 5.5. Influence of Abstraction Level Choice on Topology Generation

The abstraction level of the topologies determines the level of detail of the topologies on the highest level. The chosen abstraction level does affect the number of different components in a topology defined by its system's boundary. The level of expressiveness (component or port level) determines the level of detail on a lower level. The chosen abstraction level depends on the desired detail (granularity) of the topologies. For example, when topologies of different hybrid electric vehicle (HEV) configurations are desired, it does not make sense to take into account the drive axles or separate wheels [1]. It makes more sense to include only transmissions, electric motors, and engines. In general, the lower the abstraction level is, the more components there are, and the higher the number of possible connections is, the bigger the search space and the longer the solving time will be. However, an advantage is that topologies contain more details about the principles and connections used. Therefore, for every project, the session library needs to be filled with the problem's 'core' components. For instance, when there is a focus on the transmission topologies and the engine is used as the power source, the fuel tank is assumed to be included with the engine. The same applies for the drive axles, and the differential and the wheels are combined in the component 'wheels'. Finally, virtual nodes (VNs) are introduced, enabling the connection of different components with each other. Virtual nodes are necessary due to the components' fixed number of possible connections. By introducing VNs, this problem is solved, but they do not add any functionality. However, due to the introduction of these nodes, more isomorphous topologies are generated (cf. Section 2.4).

## 5.6. Proposed Library Format

The library format is adopted from [4] and adjusted for the purpose of our case study later on: generating an electric–hydraulic actuation system for a CVT. In the supposed library format, every component has an entry. In this entry, the category and the component type can be defined. Categories are introduced to make the engineering formalization more generic. This is useful for multi-level topology generation; a full library of constraints can be applied to a session library. Hence, the formalized engineering knowledge can be applied to every system (level). For instance, a constraint can be applied to a certain category like 'pumps'. When more pumps are defined in the session library, the constraint is applied to all those pumps. Furthermore, four different component types are distinguished: sources, sinks, converters, and virtual nodes. These types are introduced to characterize the component behavior. Sources and sinks are the system boundaries. Converters change the power flow from one to another (energy) domain, as described in the paragraph below. Virtual nodes are part of the abstraction level. These types are also used during post-processing, as described in Section 2.4. Finally, they can be used to apply constraints to all of the components belonging to a component type.

## 5.7. Component Port Definition

For every component, several ports can be defined. Next, for these ports, several properties can be set, such as specification (input or output), type (flow or signal), and energy domain (hydraulic, mechanical, electrical). For every energy, certain other properties can be set. These properties of the port and the flow properties can be used to generate constraints; however, this has not been implemented yet.

## 6. Automated Multi-Level Topology Synthesis

During this research, a software application was developed that enables multi-level topology generation for hybrid and battery-electric powertrain systems (including CVT systems). With this proposed method, feasible system topologies, including subsystems, can be automatically generated. This process is illustrated in Figure 10 with the related work at other system levels.

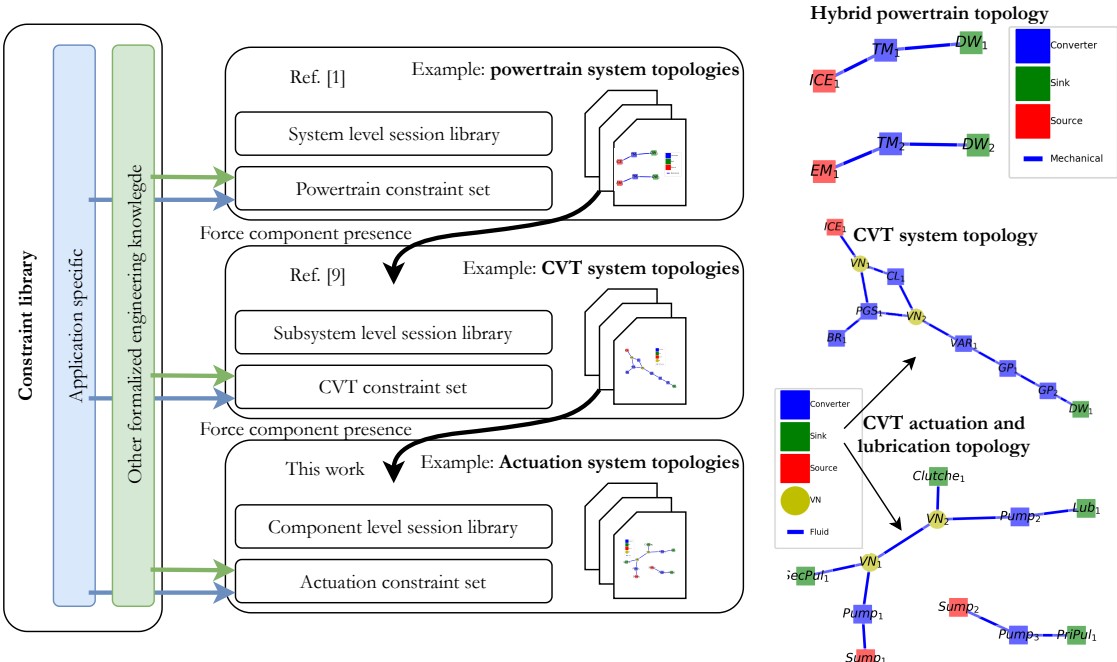

**Figure 10.** Schematic diagram of the multi-level topology generation process. Arbitrary example topologies are provided per system level. Related work is also indicated.

As can be seen, for every system and/or system level, first, a 'session library' must be composed. Secondly, for every system level, a 'set of constraints' must be composed. These can be stored in a 'central constraint library' (or 'base library'). In this base library, a complete set of 'formalized engineering knowledge' is stored. These constraints are valid at every system level. 'Application-specific constraints' might be valid for specific system levels. As illustrated in Figure 10, the generation of the lower system level is affected by the higher level due to constraint consistency. If, for example, the CVT topology contains four clutches, then the actuation system should also have that number of clutches as consumers. Therefore, the components used in a higher system level might be the system boundaries in a lower system level, as in nested graph structures [9].

### 6.1. New CAE Tool for Dynamic System Topology Synthesis

The 'session library' as part of an arbitrary 'project setup' defined by a 'system engineer' using the newly developed CAE software application is also shown in Figure 11.

The main attributes (as building blocks: 'problem definition'; 'project setup'; 'problem solving'; 'post-processing'; 'visualization'; 'model-based evaluation') of the software application developed for the 'engineer' are schematically shown (for some selection options) on the left-hand and right-hand sides, which facilitates any modifications and problem setup changes very easily and efficiently, whereas the CP solver and post-processor running on the 'software application' are shown in the middle. The sequential process steps that are in correspondence with the steps 1–5 of Section 2—yet are here realized without loss of generality for automotive powertrain systems—are indicated by the arrows.

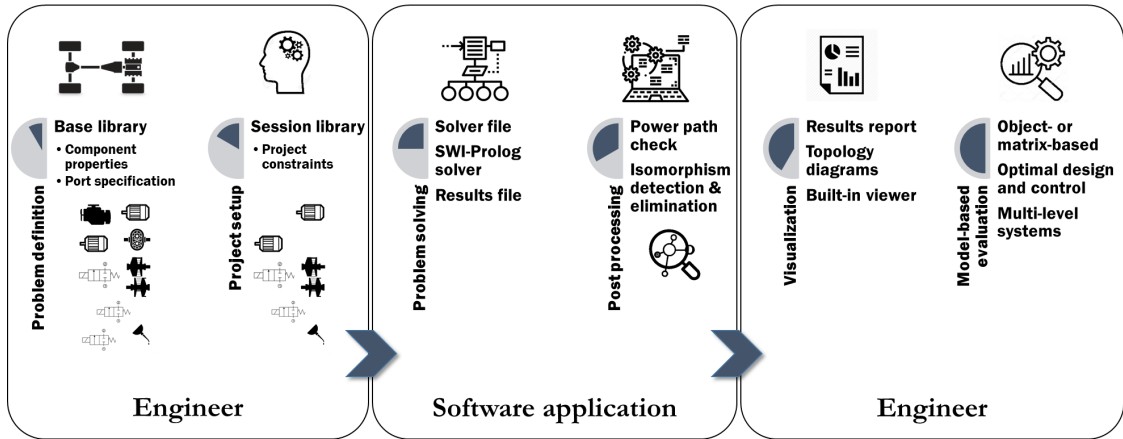

**Figure 11.** A new computer-aided engineering (CAE) tool for fully automated dynamic system topology synthesis applied to all-electric and hybrid-electric powertrain systems from the system to component level. A graphical user interface was developed for the 'engineer' based on the software attributes shown on the left-hand and right-hand sides, whereas the constraint programming (CP) solver and post-processor running on the software application are shown in the middle. The sequential process steps are indicated by the arrows.

## 7. Mechatronic Actuation Systems for CVTs

The software application developed for topology generation in the automotive field from system (powertrain topologies [1]) and subsystem (CVT topologies [9]) to the component level (actuator topologies; this work), as in nested graph structures, was analyzed using a case study (cf. Figure 10). The original algorithms, developed constraints, and processes developed in [1,9] are now fully integrated in a flexible and easily accessible software tool for system engineers, which creates a unique and novel computer-aided engineering tool in this automotive field, yet with high potential for other engineering domains as well.

### 7.1. Problem Definition: Actuation System Design

The actuation system is a part of the hydraulic system inside a CVT and serves different aspects: In addition to the actuation of the variator (clamping and speed ratio adjustments) and clutches, the hydraulic system is needed for lubrication and cooling. In conventional CVT systems, a combustion engine typically powers the oil pump of the hydraulic system. Since the oil pump is directly connected to the engine speed, the oil pump is sized according to the oil demand at idling engine speed. One of the most worst-case scenarios is maximum shifting speed when idling (harsh braking scenario), which requires a high oil flow, as well as a high pressure. As a result, during normal driving, much energy is lost, since there is a mismatch between the oil delivery and demand. Moreover, in conventional CVTs, manufacturers use the power mismatch between the oil consumers and the oil pump due to the different individual pressure and oil flow requirements of the consumers. For example, wet-plate clutches typically operate in the range of 10 to 20 bar, whereas the variator operates up to 80 bar. The efficiency of the hydraulic system can be increased by supplying an oil flow on demand (only when shifting occurs [19]) to all its consumers.

### 7.2. Project Setup: Session Library

The introduction of multiple oil pumps and the introduction of four separate sinks increases the number of design variations significantly, i.e., with every added component, the search space increases exponentially. Here, a case study is performed that optimizes the actuation system topologies for a CVT as part of a multi-level system design that is illustrated in Figure 10. For this specific problem, the library of components can be found in Table 5, and it consists of 14 components in total. For this

specific case study, the pump represents an electric oil pump that can deliver and regenerate. The flow and pressure demands of the four consumers are obtained from a CVT model according to the topology description. The component 'clutches' represents the lumped flow and pressure demand of the clutch $CL_1$ and brake $BR_1$ from the CVT system. The lubrication block represents the cooling and lubrication of the clutch, brake, push belt, gears, and bearings. Please note that geometric optimization of the pulley pistons and the variation ratio optimization are outside the scope of this project. The interested reader is referred to [20].

**Table 5.** Session library: actuation topology design synthesis for a CVT.

| Identifier | Component Type, $\tau$ | Component Name | Component Specification | Number of Instances, $\sum \iota$ | Energy Domain | |
|---|---|---|---|---|---|---|
| | | | | | Input | Output |
| A,B,C | 1 | Sump | Source | 3 | - | Hydraulic |
| D,E,F | 2 | Pump | Converter | 3 | Hydraulic | Hydraulic |
| G,H,I,J | 3 | VN | Virtual Node | 4 | Hydraulic | Hydraulic |
| K | 4 | Primary pulley | Sink | 1 | Hydraulic | - |
| L | 5 | Secondary pulley | Sink | 1 | Hydraulic | - |
| M | 6 | Lubrication | Sink | 1 | Hydraulic | - |
| N | 7 | Clutches | Sink | 1 | Hydraulic | - |

### 7.3. Project Setup: Project Constraints

Three general constraints are implemented: (i) prevention of the multiple instance redundancy; (ii) splitting of the energy domains; and, finally, (iii) the constraint of forcing the presence of components of the sink category. The following functional constraints represent the specific engineering knowledge for this case study:

- Prevention of a connection between the sink components;
- The looping of three or more virtual nodes is prevented;
- Prevention of two or three pumps in a parallel layout;
- No connection is allowed between sump and consumers; basically, bypassing of the pump is not allowed;
- No pumps in series connection, since this can be better replaced by a single pump;
- Only one sump may be connected to the pump in order to prevent oil recirculation.

### 7.4. Design Space Analysis and Model-Based Evaluation

The component expressiveness level is chosen based on the need for relatively fast exploration of the search space. When the constraint set becomes more complete, a higher expressiveness level can be chosen for topologies with more detail. The initial search space applied to 105 (14 component and 91 connection) variables results in a search space of $2^{105} \approx 4.1 \times 10^{31}$ combinations that need to be processed. Due to internal optimizations by the solver (automated evaluation of the constraints), this results in a reduced search space of $2^{59} \approx 5.8 \times 10^{17}$ combinations that need to be processed. The solving was performed in 19 min, and results were obtained for 13,054 solutions; post-processing resulted in 132 unique and feasible topologies.

When the solutions are post-processed, a complexity analysis can be performed. This analysis is to get insight into the appearance of the individual components in the feasible designs. In Figure 12a, the appearances of the number of component instances per component are provided; e.g., the number of topologies that contain two oil pumps equals 21, and the number of topologies containing three oil pumps equals 110. From that graph, it can be observed that only one topology exists that contains a single oil pump. Specifically, that topology interconnects all consumers directly with the oil pump.

In Figure 12b, as a result of a model-based evaluation step, the simulated average and maximum needed power are shown with respect to the Worldwide Harmonized Light Vehicle Test Procedure

(WLTP) [21] for a particular vehicle. Additionally, the benefits of adding one or two oil pumps to a conventional actuation topology equipped with a single pump system are clearly observable in terms of energy consumption and maximum hydraulic power over the drive cycle. Please note that this maximum power is a measure for pump dimensioning and cost, whereas the hydraulic energy saving lowers operational cost over the product's lifetime. Using this evaluation for this set of feasible topologies, the optimal architectural design can be quickly chosen or compared with the other solutions, showing the strength of the methodology.

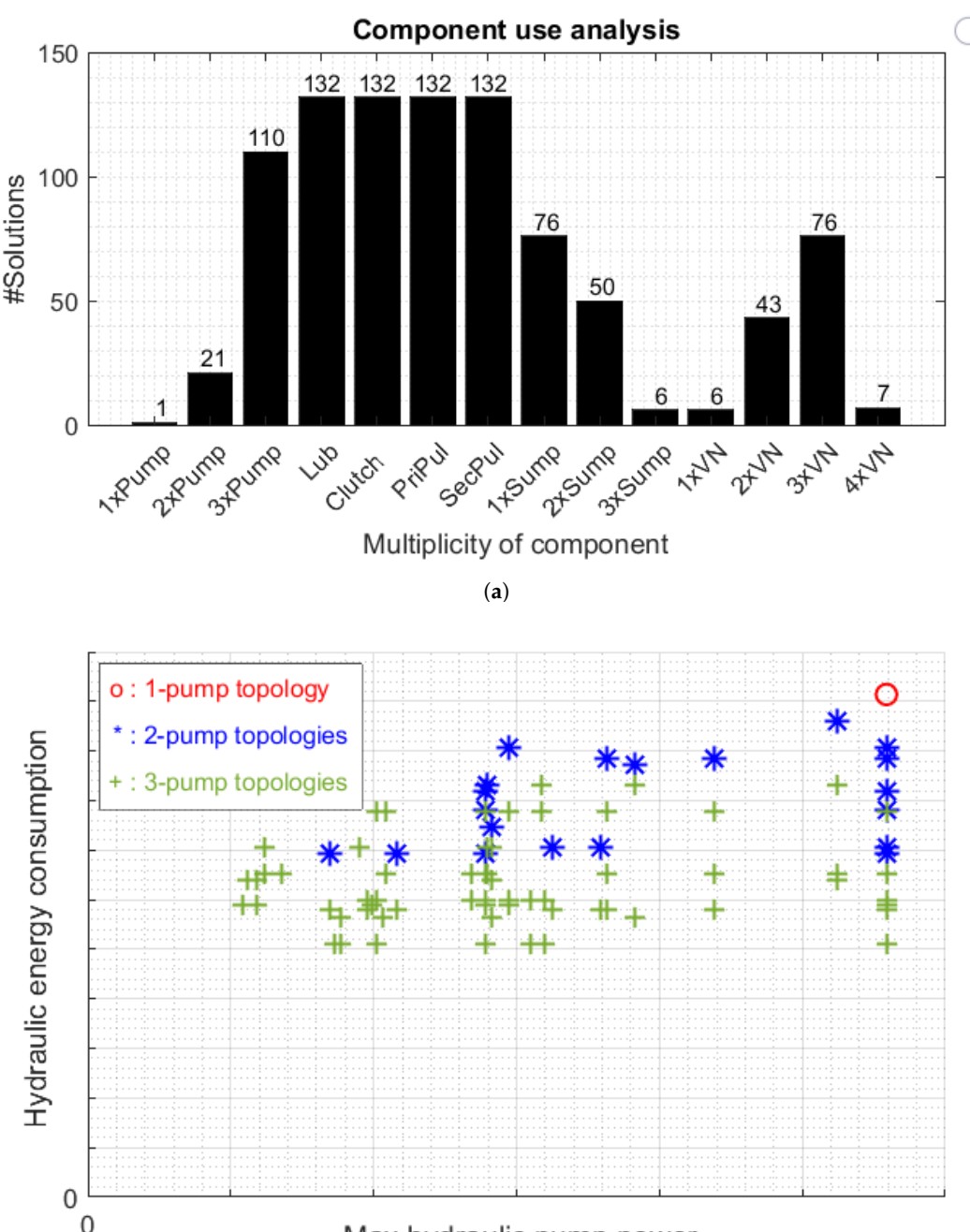

(a)

(b)

**Figure 12.** Topology generation and performance results. For reasons of confidentiality, the absolute power values are not depicted. (**a**) Histogram of the generated feasible topologies for the electro-hydraulic actuation system. (**b**) Diagram showing the average and maximum actuation power of novel and newly synthesized on-demand CVT actuation systems simulated using the Worldwide Harmonized Light Vehicle Test Procedure (WLTP) for a passenger car.

## 8. Conclusions and Recommendations

This work provides an extension and analysis of an automated method to generate discrete dynamic system topologies over multiple system levels. In contrast to other studies, the search space can easily be edited by changing the selection of session library components. The generation of constraints is fully automated based on the chosen session library components and formalized engineering knowledge. Moreover, it is possible to formalize engineering knowledge as generically as possible. This knowledge is stored and can be applied to any set of components. Additionally, the design space size and the way it is processed by the CSP solving algorithm are considered. Moreover, different levels of expressiveness are analyzed with respect to the generated topologies. The influence of the component abstraction on the search space is discussed. Finally, a case study is performed on the topology generation for electrical–hydraulic actuated subsystems as a part of a CVT system. The set of feasible topologies is further analyzed using a model-based evaluation step. To do this analyses, a new CAE software application was developed to support the generation, analysis, and evaluation steps. Future research may include: (i) further reduction of the computation time by solving the CSP. Currently, the solving is performed by SWI-Prolog and the CLP(FD) library, yet this can be improved by implementing multi-core support for this solver or by selecting another one. Furthermore, the study of different SAT solvers may lead to a reduction in solving time as well. Future research may also include (ii) an automated physical system and optimal control design for arbitrary active dynamical systems. This enables quantitative ranking of the automatically generated feasible topology designs over all three system levels.

**Author Contributions:** Conceptualization, J.W. and T.H.; Data curation, J.W.; Investigation, A.-J.K. and J.W.; Methodology, A.-J.K., J.W. and T.H.; Software, A.-J.K., J.W. and A.S.; Supervision, T.H.; Writing—original draft, A.-J.K.; Writing—review and editing, A.S. and T.H. All authors have read and agreed to the published version of the manuscript.

**Funding:** This research received no external funding.

**Conflicts of Interest:** The authors declare no conflict of interest.

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
