# Peer review of "Automated Multi-Level Dynamic System Topology Design Synthesis"

_vehicles, doi:10.3390/vehicles2040035_

Round 1

Reviewer 1 Report

The paper is well written and provides a detailed overview of a framework that can be used to design new mechatronic systems.

I'm not an expert in the field, but I had more the feeling that I was reading a report or a scientific manual than a scientific paper. I really liked the CVT example coming back throughout the text, although that made the added value of section 7 rather low.

The introduction is rather short and I would expect that there is more work available on such a topic. A 1-minute search on Google revealed, for example, these 2 papers that are potentially relevant for this work:

Königseder, Corinna, and Kristina Shea. "Visualizing relations between grammar rules, objectives, and search space exploration in grammar-based computational design synthesis." Journal of Mechanical Design 138.10 (2016).

Chenouard, Raphael, et al. "Computational design synthesis using model-driven engineering and constraint programming." Federation of International Conferences on Software Technologies: Applications and Foundations. Springer, Cham, 2016.

Small typo on Fig. 12 (a) "Clutche" -> "Clutch"

Author Response

Dear Reviewer,

Thank you very much for your useful comments and feedback. Please find our answers briefly described below.

The paper is well written and provides a detailed overview of a framework that can be used to design new mechatronic systems.

>> Thank you very much for your positive comment.

I'm not an expert in the field, but I had more the feeling that I was reading a report or a scientific manual than a scientific paper. I really liked the CVT example coming back throughout the text, although that made the added value of section 7 rather low.

>> Yes, we put also emphasize on the methodology development and used the CVT as a case study that can be seen a part of the development of larger simulation design tool. The results presented in this work contributes to the work done on system architecture design at a higher system level (powertrain). We made a reference to that aspect.

The introduction is rather short and I would expect that there is more work available on such a topic. A 1-minute search on Google revealed, for example, these 2 papers that are potentially relevant for this work:

Königseder, Corinna, and Kristina Shea. "Visualizing relations between grammar rules, objectives, and search space exploration in grammar-based computational design synthesis." Journal of Mechanical Design 138.10 (2016).

Chenouard, Raphael, et al. "Computational design synthesis using model-driven engineering and constraint programming." Federation of International Conferences on Software Technologies: Applications and Foundations. Springer, Cham, 2016.

>> Yes, we fully agree – so we decided to follow your advice to extent the introduction and add more references.

Small typo on Fig. 12 (a) "Clutche" -> "Clutch"

>> Corrected.

Yours sincerely,

Theo Hofman (on behalf of the other co-authors.)

Reviewer 2 Report

well-structured work. the parties are connected to each other with sufficient scientific rigour. the simulation part is described in detail as well as the method used and the steps of the procedure. maybe only the bibliography is a little limited and could be expanded

Author Response

Dear Reviewer,

Thank you very much for the positive feedback. We added some additional references on your advice.

Yours sincerely,

Theo Hofman (on behalf of the other co-authors)